Genome-wide identification, phylogenetic and expression pattern analysis of Dof transcription factors in blueberry (Vaccinium corymbosum L.)

Li Tianjie 1
Wang Xiaoyu 2
Elango Dinakaran 3
Zhang Weihua 1
Li Min 2
Zhang Fan 1
Pan Qi 1
Wu Ying wuying35969@126.com 1
1 Tianjin Agricultural University , Tianjin , China
2 Inner Mongolia Minzu University , Mongolia , China
3 Iowa State University , Ames , United States of America
Uversky Vladimir
Electronic publication date: 2022 Oct 3
Publication date: 2022
Volume: 10
Electronic Location ID: e14087
Received 2022 Jul 6; Accepted 2022 Aug 29
Copyright: ©2022 Li et al.
Copyright year: 2022
Copyright holder: Li et al.
License: This is an open access article distributed under the terms of the Creative Commons Attribution License, which permits unrestricted use, distribution, reproduction and adaptation in any medium and for any purpose provided that it is properly attributed. For attribution, the original author(s), title, publication source (PeerJ) and either DOI or URL of the article must be cited.
License URL: https://creativecommons.org/licenses/by/4.0/

Keywords: Dof transcription factor, Phylogeny, Expression pattern analysis, Blueberry

Funding: National Natural Science Foundation of China 61971312 Postgraduate Research and Innovation Funding Project of Tianjin Agricultural University 2021XY038 Special Project for the Introduction of Leading Talents in the Disciplines of Higher Education Institutions in Tianjin of China This work was supported by the National Natural Science Foundation of China (No. 61971312), the Postgraduate Research and Innovation Funding Project of Tianjin Agricultural University (No. 2021XY038), the Special Project for the Introduction of Leading Talents in the Disciplines of Higher Education Institutions in Tianjin of China. The funders had no role in study design, data collection and analysis, decision to publish, or preparation of the manuscript.

==============================
Background

DNA binding with one finger (Dof) proteins are plant-specific transcription factor (TF) that plays a significant role in various biological processes such as plant growth and development, hormone regulation, and resistance to abiotic stress. The Dof genes have been identified and reported in multiple plants, but so far, the whole genome identification and analysis of Dof transcription factors in blueberry (Vaccinium corymbosum L.) have not been reported yet.

Methods

Using the Vaccinium genome, we have identified 51 VcDof genes in blueberry. We have further analyzed their physicochemical properties, phylogenetic relationships, gene structure, collinear analysis, selective evolutionary pressure, cis-acting promoter elements, and tissue and abiotic stress expression patterns.

Results

Fifty-one VcDof genes were divided into eight subfamilies, and the genes in each subfamily contained similar gene structure and motif ordering. A total of 24 pairs of colinear genes were screened; VcDof genes expanded mainly due to whole-genome duplication, which was subjected to strong purifying selection pressure during the evolution. The promoter of VcDof genes contains three types of cis-acting elements for plant growth and development, phytohormone and stress defense responsiveness. Expression profiles of VcDof genes in different tissues and fruit developmental stages of blueberry indicated that VcDof2 and VcDof45 might play a specific role in anthesis and fruit growth and development. Expression profiles of VcDof genes in different stress indicated that VcDof1, VcDof11, and VcDof15 were highly sensitive to abiotic stress. This study provides a theoretical basis for further clarifying the biological function of Dof genes in blueberry.

Introduction

Blueberry is a perennial woody plant that belongs to the genus Vaccinium. It is one of the five healthy fruits recommended by the Food and Agriculture Organization of the United Nations; it is rich in anthocyanins, vitamins, flavonoids, and other active ingredients (Kalt et al., 2020; Li et al., 2020). It has the functions of protecting eyesight, preventing type 2 diabetes, anti-aging, and anti-cancer (Ma et al., 2018; Ono-Moore et al., 2016; Istek & Gurbuz, 2017). Due to various health benefits of blueberries and awareness of healthy diets, consumption of blueberries and their derivatives increased tremendously. So far, blueberry has grown in 71 countries (Gallardo et al., 2018). Rapidly changing climatic conditions may drastically affect the global production of blueberries in the coming years. So, understanding and identifying genetic factors’ response to biotic and abiotic stress tolerance is imperative. Research shows that salt and drought stress alone in blueberry causes 25–30% yield losses (Wang et al., 2021a; Wang et al., 2021b).

A transcription factor (TF) is a DNA binding protein that can specifically interact with cis-acting elements of genes and regulate the specific expression of target genes (Zhang & Hou, 2021). Understanding its structure and function is very important to recognize the biological role of the gene in the given context (Chen et al., 2017). DNA binding with one finger (Dof) TFs consist of 200–400 amino acids (Gupta et al., 2015). Dof includes the DNA-binding domain and domain involved in transcriptional activation or repression and exhibits regulatory roles in nuclei (Chattha et al., 2020; Liu et al., 2020). But, except for the highly conserved domain of Dof, the amino acid sequences are not well conserved, this creates a greater diversity across family members, and individual proteins exhibit very divergent sequences (Wu et al., 2016; Cominelli et al., 2011; Yang et al., 2011), which may be an important reason for the functional diversity of Dof proteins and involved in varied plant physiological and biochemical processes (Yanagisawa, 2002).

In 1993, the first Dof transcription factor ZmDof1 was identified and cloned in maize (Yanagisawa & Izui, 1993). With the recent developments in genome sequencing and bioinformatics capabilities, multiple Dof TFs have been identified in Arabidopsis thaliana, Solanum tuberosum, Solanum lycopersicum, Oryza sativa, and Zea mays (Lijavetzky, Carbonero & Vicente-Carbajosa, 2003; Venkatesh & Park, 2015; Cai et al., 2013; Jiang et al., 2012; Moreno-Risueno et al., 2007). Dof TFs have been shown to play an important role in plant growth and development, primary and secondary metabolism, hormone regulation, and abiotic stress resistance (Renau-Morata et al., 2017; Yanagisawa, 2004; Skirycz et al., 2007; Li et al., 2022). Specifically, inducible overexpression of AtDof5.4/OBP4 in Arabidopsis promoted early endocycle onset, inhibited cell expansion, and reduced cell size and number, resulting in dwarf plants (Xu et al., 2016). AtDof4.7 regulates the expression of cell wall hydrolysis enzymes; overexpression of AtDof4.7 causes an abscission-related polygalacturonase gene PGAZAT down-regulation, which affects the shedding of flower organs (Wei et al., 2010). AtDof6 can regulate seed germination by interacting with TCP14 protein and affecting ABA anabolism (Rueda-Romero et al., 2012). Overexpression of GmDof4 and GmDof11 genes in Glycine max increased the content of total fatty acids and lipids of transgenic Arabidopsis seeds (Wang et al., 2007). Overexpression of tomato Dof gene family member TDDF1 in plants can improve tomato tolerance to drought, salt, and various hormone treatments (Ewas et al., 2017). In Tamarix hispida, ThDof1.4 enhances proline levels, reactive oxygen species (ROS) scavenging capacity, and tolerance to salt and osmotic stress (Zang et al., 2017). Many studies have shown that the Dof transcription factor gene family plays an important role in plant growth, development, and resistance to abiotic stress.

The haplotype-phased genome assembly of blueberry was published in 2019 (Colle et al., 2019). So far, there are few reports on the role of Dof TFs in abiotic stress tolerance. Therefore, we aim to identify Dof TFs in blueberry using the whole genome of blueberry, and we further analyzed gene structure, phylogenetic relationships, collinear analysis, and the expression pattern of VcDof genes under abiotic stress in different tissue types was analyzed using RNA-Seq and qRT-PCR. In summary, the present research provides important information on the potential function of the blueberry Dof TFs in abiotic stress tolerance.

Materials & Methods

Identification and characterization of Dof transcription factors in blueberry

The blueberry genome and annotation data were downloaded using the Vaccinium database (GDV, https://www.vaccinium.org). The Dof TFs protein sequences of Arabidopsis and rice were used as query sequences (https://phytozome-next.jgi.doe.gov/). The blueberry genome was compared by Basic Local Alignment Search Tool Protein (BLASTP) in a Linux system, and the screening threshold E-value was 1e−5. The protein sequences of candidate genes were obtained. Furthermore, the Dof domain, PF02701, was used to search the blueberry genome using HMMER software (Potter et al., 2018). By integrating the results of the above two steps, the sequences of these genes were submitted to SMART (http://smart.embl.de/smart/batch.pl) and NCBI-CDD (https://www.ncbi.nlm.nih.gov/cdd/) to remove redundant and non-conservative genes (Letunic & Bork, 2018; Lu et al., 2020). Then, the gene sequences of blueberry Dof TFs were obtained, named according to their position on the chromosome scaffold.

The amino acid number, theoretical isoelectric point (pI), and molecular weight (MW) of VcDof genes were analyzed online using the ProtParam tool (https://web.expasy.org/protparam). Subcellular localization information of VcDof genes was predicted using CELLO (http://cello.life.nctu.edu.tw) and WoLF PSORT (https://wolfpsort.hgc.jp).

Multiple sequence alignments and phylogenetic analysis

The protein sequences of Dof TFs in Arabidopsis thaliana and Oryza sativa were downloaded from NCBI (https://www.ncbi.nlm.nih.gov/) and phytozome (https://phytozome-next.jgi.doe.gov/). Multiple sequence alignments (MSA) were performed using MegaX software (Kumar et al., 2018). Based on multiple sequence alignment results, a phylogenetic tree was constructed using the maximum likelihood (ML) method; the optimal fitting model was selected as JTT+G, and the number of Bootstrap tests was adjusted to 1000. The resulting phylogenetic tree file was enhanced using iTol (https://itol.embl.de/, Letunic & Bork, 2021).

Gene structure analysis of the Dof transcription factors

Based on the blueberry genome structure annotation information file, the number and location of exons/introns of VcDof genes were counted by Gene Structure Display Server (GSDS2.0, http://gsds.gao-lab.org, Hu et al., 2015a; Hu et al., 2015b). The conservative motif of blueberry Dof protein was analyzed online by Maximization for Motif Elicitation program (MEME, https://meme-suite.org/meme/tools/meme, Bailey et al., 2015). The TBtools were used for visualization (Chen et al., 2020).

Collinearity analysis of Dof transcription factors in blueberry

Duplication events of VcDof genes were analyzed by the MCscanX (Tang et al., 2008) and visualized in TBtools. Using Ka/Ks ratio to display evolutionary selection pressure between collinearity gene pairs, Ka/Ks>1, = 1, <1 indicated positive selection, neutral evolution, and purifying selection, respectively. Further, T = Ks/2 λ (T: calculates divergence time; Mya: million years; λ: replacement rate, λ = 1.3 × 10−8) was used to compute the approximate date of duplication and divergence events (Colle et al., 2019).

Promoter cis-acting element analysis

The 2,000 bp promoter region upstream of VcDof genes were obtained from the blueberry genome, and the cis-acting elements in the promoter region were analyzed using the tool PlantCARE (http://bioinformatics.psb.ugent.be/webtools/plantcare/html).

Expression profile of Dof transcription factors in different tissues

Using the RNA-Seq data, the expression level of VcDof genes in different tissues and fruit development stages were identified (https://www.vaccinium.org/; NCBI accession number: PRJNA494180). The transcript abundance was estimated using FPKM (fragments per kilobase per million measure) after log2 conversion (Colle et al., 2019). Hierarchical clustered heatmaps and visualization were done using the TBtools.

Plant material and experimental treatments

In this study, the saplings of the northern highbush blueberry ‘Bluecrop’ preserved in the laboratory were used as experimental materials, and the pots were filled with nutrient soil and vermiculite (volume ratio 1:1). The effective components of nutrient soil were nitrogen 140 mg/kg−1, phosphorus 100 mg/kg−1, potassium 180 mg/kg−1, organic matter content 91.3%, PH 5.5. The biennial sapling plants were placed in the greenhouse of Tianjin Agricultural University. The deionized water and 1/2 Hoagland nutrient solution were regularly irrigated to make the plants grow under the optimal growth conditions.

This experimental design was a completely randomized design with three treatments. The specific treatments were as follows: 150 mMol/L NaCl was used to irrigate blueberry plants to simulate salt stress; 20% PEG6000 was used to irrigate the plants to simulate drought stress; and ABA hormone 100 µMol/L was used to spray blueberry leaves to simulate adversity stress. After 0(control), 3, 6, 12, and 24 h of each treatment, leaf samples were collected. Triplicate leaf samples were collected for each time point and treatment and frozen in liquid nitrogen. The samples were stored in an ultra-low temperature freezer at −80 °C (Bano et al., 2021; Han, Wu & Li, 2021).

RNA isolation, cDNA preparation, and quantitative RT-PCR analysis

The RNA extraction was done using the Easypure Plant RNA Kit (TransGen Biotech, China). After gel electrophoresis detected clear RNA bands and apparent separation, cDNA was synthesized using the kit PrimeScript TMRT Master Mix (TaKaRa BIO INC., Japan). The blueberry EIF (VcEIF) gene was used as the reference (Deng, Li & Sun, 2020), and the specific primers for the target gene and the reference gene were designed using Premier 5.0 software. The instrument used for qRT-PCR was qTOWER 2.2 (Analytik Jena, Germany), and the qRT-PCR reaction was performed using the iTaq Universal SYBR® Green Supermix (Bio-Rad INC., USA). The relative expression levels were calculated as 2 −ΔΔCt (Livak & Schmittgen, 2001), the significance of the difference between each treatment group compared to control was determined using one-way ANOVA in SPSS 26.0 (SPSS Inc., USA), error bars indicate standard deviation, and asterisks indicate significant differences between the treatments and control, ∗p ≤ 0.05, ∗∗p ≤ 0.01, ∗∗∗p ≤ 0.001.

Results

Genome-wide identification of Dof transcription factors in blueberry

The 51 blueberry Dof TFs were identified and named VcDof1 to VcDof51 according to the position of the genes on the chromosome scaffold (Table 1). The length of amino acids varied greatly, with VcDof40 the longest, containing 493 amino acids, and VcDof36 the shortest, containing 118 amino acids. The analysis of physicochemical properties using ProtParam showed that the molecular weight of VcDof genes was between 13,621.8 and 53,701.66 Da. The theoretical isoelectric point was 4.56–10.56 for both acidic and alkaline proteins. The theoretical isoelectric point of VcDof15 is the smallest, showing a higher precipitation coefficient. The theoretical isoelectric point of VcDof36 is the largest, suggesting that it has strong solubility and weak precipitation ability. Subcellular localization prediction showed that most VcDof genes were located in the nucleus.

Table 1 The basic information of Dof TFs in blueberry.

Gene name	Gene ID	Chromosome location	CDS (bp)	Length (aa)	pI	Molecular weight (Da)	Subcellular location	
VcDof1	VaccDscaff3-snap-gene-159.14	VaccDscaff3:15954245-15956961	1200	399	8.31	44,373.94	Nuclear	
VcDof2	VaccDscaff4-augustus-gene-345.25	VaccDscaff4:34550915-34552763	876	291	8.1	31,800.02	Nuclear	
VcDof3	VaccDscaff8-augustus-gene-261.30	VaccDscaff8:26158034-26160021	918	305	9.26	33,049.91	Nuclear	
VcDof4	VaccDscaff9-augustus-gene-161.15	VaccDscaff9:16127607-16130475	801	266	9.18	29,743.08	Nuclear	
VcDof5	VaccDscaff10-snap-gene-247.18	VaccDscaff10:24688121-24689836	906	301	9.26	32,614.39	Nuclear	
VcDof6	VaccDscaff11-processed-gene-136.2	VaccDscaff11:13655066-13655560	495	164	9.43	18,177.54	Nuclear	
VcDof7	VaccDscaff11-snap-gene-158.12	VaccDscaff11:15805973-15808315	1071	356	8.96	39,331.62	Nuclear	
VcDof8	VaccDscaff11-processed-gene-168.11	VaccDscaff11:16838900-16843433	612	203	9.85	22,095.71	Nuclear	
VcDof9	VaccDscaff11-processed-gene-307.12	VaccDscaff11:30754783-30756949	951	316	7.63	34,553.12	Nuclear	
VcDof10	VaccDscaff11-augustus-gene-330.45	VaccDscaff11:33025388-33027124	867	288	9.43	30,770.94	Nuclear	
VcDof11	VaccDscaff12-processed-gene-81.3	VaccDscaff12:8170002-8170682	681	226	6.37	24,042.46	Nuclear	
VcDof12	VaccDscaff13-augustus-gene-79.30	VaccDscaff13:7890522-7894462	1470	489	5.74	53,351.37	Nuclear	
VcDof13	VaccDscaff13-augustus-gene-135.30	VaccDscaff13:13508244-13511349	957	318	9.04	34,095.62	Nuclear	
VcDof14	VaccDscaff13-augustus-gene-254.20	VaccDscaff13:25461888-25465267	1407	468	5.57	51,450.94	Nuclear	
VcDof15	VaccDscaff14-processed-gene-266.8	VaccDscaff14:26675412-26676173	762	253	4.56	28,167.22	Nuclear	
VcDof16	VaccDscaff15-snap-gene-139.15	VaccDscaff15:13899895-13902343	1062	353	9.07	39,109.51	Nuclear	
VcDof17	VaccDscaff15-processed-gene-145.13	VaccDscaff15:14528524-14530344	795	264	9.3	28,571.85	Nuclear	
VcDof18	VaccDscaff15-augustus-gene-292.25	VaccDscaff15:29225641-29227947	954	317	7.21	34,696.27	Nuclear	
VcDof19	VaccDscaff15-processed-gene-320.11	VaccDscaff15:32030644-32032478	1281	426	9.88	46,664.59	Nuclear	
VcDof20	VaccDscaff16-augustus-gene-64.37	VaccDscaff16:6408947-6412982	1395	464	5.93	50,683.97	Nuclear	
VcDof21	VaccDscaff17-augustus-gene-280.24	VaccDscaff17:28047716-28050042	963	320	6.52	35,155.14	Nuclear	
VcDof22	VaccDscaff17-processed-gene-380.10	VaccDscaff17:38033194-38035650	885	294	8.11	32,152.67	Nuclear	
VcDof23	VaccDscaff19-augustus-gene-7.37	VaccDscaff19:722925-724763	873	290	8.81	32,131.57	Nuclear	
VcDof24	VaccDscaff19-processed-gene-223.3	VaccDscaff19:22351441-22352415	975	324	9.20	35,667.63	Nuclear	
VcDof25	VaccDscaff19-processed-gene-236.11	VaccDscaff19:23614873-23616681	789	262	9.30	28,419.66	Nuclear	
VcDof26	VaccDscaff19-snap-gene-338.49	VaccDscaff19:33815798-33819654	1182	393	6.00	43,431.08	Nuclear	
VcDof27	VaccDscaff19-augustus-gene-349.24	VaccDscaff19:34887825-34889658	1239	412	9.93	45,154.72	Nuclear	
VcDof28	VaccDscaff20-augustus-gene-371.28	VaccDscaff20:37159924-37162529	990	329	8.13	35,787.13	Nuclear	
VcDof29	VaccDscaff21-processed-gene-152.13	VaccDscaff21:15189887-15192636	1014	337	8.91	36,226.51	Nuclear	
VcDof30	VaccDscaff21-augustus-gene-306.24	VaccDscaff21:30639419-30644218	1245	414	6.47	45,615.07	Nuclear	
VcDof31	VaccDscaff22-augustus-gene-267.28	VaccDscaff22:26762373-26763954	753	250	8.34	27,207.32	Nuclear	
VcDof32	VaccDscaff22-processed-gene-275.2	VaccDscaff22:27553163-27555116	939	312	8.81	33,950.99	Nuclear/ Peroxisome	
VcDof33	VaccDscaff24-snap-gene-94.38	VaccDscaff24:9471913-9475742	1197	398	5.76	44,173.03	Nuclear	
VcDof34	VaccDscaff24-processed-gene-229.1	VaccDscaff24:22899664-22900548	885	294	8.13	32,341.95	Nuclear	
VcDof35	VaccDscaff24-processed-gene-290.7	VaccDscaff24:29059975-29060412	438	145	10.48	16,520.86	Nuclear	
VcDof36	VaccDscaff27-snap-gene-20.34	VaccDscaff27:2024971-2025485	357	118	10.56	13,621.80	Nuclear	
VcDof37	VaccDscaff27-augustus-gene-35.18	VaccDscaff27:3516664-3518800	1026	341	9.14	35,675.48	Nuclear/ Peroxisome	
VcDof38	VaccDscaff28-augustus-gene-4.38	VaccDscaff28:457494-460214	993	330	8.13	35,901.23	Nuclear	
VcDof39	VaccDscaff29-snap-gene-153.37	VaccDscaff29:15276983-15278775	960	319	9.16	34,455.72	Nuclear	
VcDof40	VaccDscaff30-augustus-gene-326.23	VaccDscaff30:32673932-32678040	1482	493	5.83	53,701.66	Nuclear/ mitochondrial	
VcDof41	VaccDscaff31-processed-gene-100.12	VaccDscaff31:10070463-10071110	648	215	9.05	22,434.25	Nuclear	
VcDof42	VaccDscaff33-augustus-gene-185.34	VaccDscaff33:18572264-18573956	966	321	9.14	34,766.97	Nuclear	
VcDof43	VaccDscaff34-augustus-gene-312.13	VaccDscaff34:31205413-31207358	969	322	9.20	33,725.28	Nuclear/ Peroxisome	
VcDof44	VaccDscaff35-processed-gene-123.5	VaccDscaff35:12329415-12329894	480	159	9.27	17,650.15	Nuclear	
VcDof45	VaccDscaff35-augustus-gene-253.36	VaccDscaff35:25292011-25293987	879	292	8.10	31,873.07	Nuclear	
VcDof46	VaccDscaff37-augustus-gene-143.22	VaccDscaff37:14293905-14295828	837	278	9.40	30,154.54	Nuclear	
VcDof47	VaccDscaff40-processed-gene-12.4	VaccDscaff40:1275799-1276467	669	222	6.64	23,658.98	Nuclear	
VcDof48	VaccDscaff41-processed-gene-58.8	VaccDscaff41:5852050-5852715	666	221	6.64	23,614.92	Nuclear	
VcDof49	VaccDscaff46-augustus-gene-100.28	VaccDscaff46:10034713-10036487	747	248	8.34	27,093.22	Nuclear	
VcDof50	VaccDscaff46-snap-gene-129.47	VaccDscaff46:12897112-12899008	936	311	8.58	33,975.09	Nuclear/ Peroxisome	
VcDof51	VaccDscaff48-augustus-gene-102.22	VaccDscaff48:10277144-10279070	864	287	8.81	31,974.36	Nuclear	

Phylogenetic tree of the Dof transcription factors in blueberry

To further understand the evolutionary relationship of Dof TFs, the amino acid sequences of Dof genes in blueberry, Arabidopsis, and rice was compared using the MegaX software. The results showed that (Fig. 1), all Dof genes were divided into four main subfamilies (subfamilies A–D), which could be divided into multiple subfamilies, A, B1, B2, C1, C2.1, C2.2, C3, D1, and D2 with supported bootstrap values. Analysis of each subfamily found that the D1 subfamily contained the largest number of Dof TFs, consistent with Wei’s finding that the D1 subfamily had the most members in the eggplant phylogenetic tree (Wei et al., 2018). The Dof TFs of dicotyledonous blueberry, Arabidopsis, and monocotyledonous rice did not show apparent separation in the phylogenetic tree, indicating that in the long evolutionary process, Dof TFs did not appear obvious differentiation between monocotyledonous and dicotyledonous plants, may contain similar functions. Therefore, the Dof TFs are relatively conserved in the evolution of plants. Interestingly, VcDof and OsDof genes did not appear in the C3 subfamily.

Figure 1 Phylogenetic tree of the Arabidopsis, rice and blueberry Dof transcription factors.

The nine subfamilies were shown in different colors. The blue filled pentagram denoted VcDof genes; the green-filled pentagram denoted AtDof genes; the yellow-filled pentagram denoted OsDof genes.

Gene structure and conserved motifs analysis of the Dof transcription factors

In order to clarify the genetic structure of the VcDof genes, the conserved motifs of VcDof genes were analyzed by MEME. The results showed that ten motifs (motif1-motif10) were obtained (Fig. S1). Motif1 was the conserved motif of the Dof transcription factor, and motif1 was contained in each identified VcDof gene, proving the identification results reliability. The arrangement and number of motifs in each subfamily were the same (Fig. 2A). Studies have shown that the gene’s distribution of exons and introns is closely related to the evolution of the gene family. Gene structure analysis results show that VcDof genes contain coding sequence (CDS) and untranslated region (UTR) with different numbers and lengths (Fig. 2B). The number of introns in VcDof genes was 0–3, 66.7% of VcDof genes contained only one intron, VcDof4, VcDof26, and VcDof33 had three introns, and members of the same subfamily had similar numbers of introns. There are differences in intron length between different subfamilies, possibly due to the absence or increase of introns in the long-term evolution process. VcDof genes belong to the exon-poor subgroup, indicating that blueberry Dof TFs are relatively conservative in the evolutionary process (Hu et al., 2015a; Hu et al., 2015b).

Figure 2 Conserved protein motifs and gene structure analysis of Dof transcription factors in blueberry.

(A) Identified conserved protein motifs in the VcDof genes. Each motif was indicated with a specific color, different colors of lines denoted the different subfamilies; (B) Gene structure of the VcDof genes. The green box represented the CDS region and yellow box represented the UTR, and the grey lines represented introns.

Collinearity gene pair and divergence time analysis of the Dof transcription factors in blueberry

Due to the incomplete splicing and chromosomal assembly of the blueberry genome, the genes can only be located on the chromosomal scaffold of the blueberry. The results showed that (Fig. 3), 24 pairs of collinearity genes were found on 21 chromosome scaffolds, showing uneven distribution; VaccDscaff19 and VaccDscaff11 contained the most collinearity genes with four pairs. We have found only a pair of collinearity genes in the chromosome 9 scaffold. The non-synonymous/synonymous mutation ratio (Ka/Ks) is a common tool to study the selection pressure of gene evolution. This study calculated the evolutionary selection pressure of VcDof collinearity gene pairs. The results showed that the Ka/Ks ratio of 91.67% collinear gene pairs were all less than 1, subjected to purifying selection during the evolution process (Table 2), while VcDof7-VcDof24 and VcDof32-VcDof50 were subjected to positive selection. Overall, the VcDof genes were subjected to intense purifying selection pressure during the evolution process. Whole-genome duplication (WGD) events provided the main driving force for the evolution of the VcDof genes. The estimated time of divergent events between collinear genes was 0.2699–95.6425 Mya ago.

Figure 3 Chromosomal location and collinearity analysis of Dof TFs in blueberry.

The outer color ring box in the circle represented the chromosome scaffold, the color part inside the circle was the blueberry genome collinear region, and the red line was the blueberry Dof TFs collinear gene pair.

Table 2 The evolution selection pressure and divergence time of Dof TFs in blueberry.

Duplicated pair	Duplicated model	Ka	Ks	Ka/Ks	Selection pressure	Divergence time (Mya)	
VcDof1-VcDof12	WGD	0.365234001	2.486705939	0.146874625	purify selection	95.6425361	
VcDof2-VcDof45	WGD	0.001505269	0.060032031	0.02507443	purify selection	2.30892426	
VcDof3-VcDof5	WGD	0.002911212	0.033338822	0.087321976	purify selection	1.282262382	
VcDof5-VcDof27	WGD	0.359266416	1.723444937	0.208458308	purify selection	66.28634373	
VcDof5-VcDof29	WGD	0.417288029	1.517602966	0.274965217	purify selection	58.36934483	
VcDof7-VcDof24	WGD	0.013538274	0.00899561	1.504986721	positive selection	0.345985003	
VcDof8-VcDof17	WGD	0.063654913	0.144930931	0.439208611	purify selection	5.574266561	
VcDof9-VcDof18	WGD	0.001353791	0.009646435	0.140341064	purify selection	0.37101674	
VcDof10-VcDof19	WGD	0.004531736	0.025426164	0.178231206	purify selection	0.977929375	
VcDof12-VcDof20	WGD	0.149983742	0.709859291	0.21128658	purify selection	27.30228042	
VcDof12-VcDof40	WGD	0.007081265	0.040226191	0.176036186	purify selection	1.547161197	
VcDof17-VcDof25	WGD	0.005002798	0.027627433	0.181080807	purify selection	1.062593578	
VcDof18-VcDof26	WGD	0.032607364	0.036268486	0.899055024	purify selection	1.394941757	
VcDof19-VcDof27	WGD	0.003165675	0.007017595	0.451105363	purify selection	0.269907502	
VcDof20-VcDof40	WGD	0.14460691	0.708639164	0.204062825	purify selection	27.25535246	
VcDof25-VcDof34	WGD	0.006670416	0.010969117	0.608108789	purify selection	0.421889112	
VcDof26-VcDof31	WGD	0.224480814	0.565207401	0.397165382	purify selection	21.7387462	
VcDof27-VcDof32	WGD	0.293672643	0.906127367	0.324096428	purify selection	34.85105259	
VcDof28-VcDof38	WGD	0.001305483	0.018377642	0.071036498	purify selection	0.706832396	
VcDof29-VcDof39	WGD	0.018104628	0.036550818	0.495327557	purify selection	1.405800703	
VcDof32-VcDof50	WGD	0.025707622	0.009371461	2.743181808	positive selection	0.360440817	
VcDof33-VcDof49	WGD	0.220819905	0.542497777	0.407042967	purify selection	20.86529913	
VcDof37-VcDof43	WGD	0.008295015	0.038690527	0.214393966	purify selection	1.488097177	
VcDof39-VcDof42	WGD	0.015991336	0.034226471	0.467221286	purify selection	1.316402732	

Promoters analysis of the Dof transcription factors in blueberry

Transcription factors bind to cis-acting elements of promoters to initiate gene transcription, and promoters are key factors that determine the spatiotemporal expression and transcription levels of genes (Brázda, Bartas & Bowater, 2021). In this study, PlantCARE was used to analyze the cis-acting promoter elements of 2,000 bp upstream of the start site of the VcDof genes (Fig. 4). All cis-acting promoters were classified into three categories according to their functions: plant growth and development, phytohormone responsiveness, and stress defense responsiveness. Plant growth and development-related elements analysis showed that the ‘as-1 promoter’ was most distributed in VcDof genes, accounting for 32%, and 18% with the ‘O2-site’ promoter related to cis-acting regulatory element involved in zein metabolism regulation. Among the phytohormone responsiveness elements, ABRE (abscisic acid responsiveness) was the most numerous promoter (30%), which was the ABA-responsive element. 86.3% of the VcDof genes promoter contained ABRE elements, suggesting that the VcDof genes played an important role in regulating abscisic acid. STRE (thermal stress responsive element) was the promoter with the highest proportion (28%) in the stress defense responsiveness elements, and it also contained TC-rich repeats (6%), LTR (long terminal repeats, 9%), and MBS (MYB binding site, 3%) promoters. Studies have shown that these promoters are involved in defense and stress tolerance (Wang et al., 2018). VcDof11 gene contains 25 abiotic stress-related promoters, one of the most stress tolerance elements in VcDof genes. It is speculated that VcDof11 plays an important role in the response of blueberry to abiotic stress.

Figure 4 The cis-acting elements of Dof TFs in blueberry.

(A) The gradient red colors indicate the number of cis-acting elements; (B) color-coded histograms indicate the number of cis-acting elements of genes in each category; (C) pie charts show the proportion of different cis-acting elements in each category.

Expression profiles of blueberry Dof transcription factors in different tissues and fruit development stages

Using the RNA-seq data, we have explored the expression profile of different tissues at different time points. A heatmap of VcDof genes for root, shoot, leaf (day and night), flower (bud, anthesis, and petal), and fruit (green, pink, and ripe) is available (Fig. 5). It was found that, except for VcDof24, VcDof35, and VcDof36, other blueberry Dof TFs were detected in various blueberry tissues. Overall, all blueberry Dof TFs were highly expressed in the root. Among them, a total of 15 VcDof genes, VcDof8, VcDof9, VcDof14, VcDof17, VcDof19, VcDof20, VcDof25, VcDof31, VcDof34, VcDof38, VcDof40, VcDof45, VcDof47, VcDof48, and VcDof49 were highly expressed in different tissues and fruit development stages of blueberry, suggesting that they may play an important positive regulatory role in blueberry growth and development. However, the expression levels of 7 VcDof genes, including VcDof1, VcDof2, VcDof5, VcDof10, VcDof13, VcDof23, and VcDof42 were relatively low in different tissues and fruit development stages of blueberry. Therefore, they may play a negative feedback regulation role in blueberry growth and fruit ripening. Interestingly, the expression level of VcDof45 gradually increased during the process of blueberry fruit ripening (green to ripe), while the expression level of VcDof2 gradually decreased. It is speculated that these two genes play important regulatory roles in the development of blueberry fruit. The remaining blueberry Dof TFs showed different expression levels in different tissues, which may have different biological functions.

Figure 5 Expression profiles of Dof TFs in different tissues and fruit development stages of blueberry.

The blue or red indicated lower or higher expression levels of each transcript in each sample, respectively.

Expression profiles of blueberry Dof transcription factors in response to salt, drought and abscisic acid

Previous studies have shown that Dof TFs are widely involved in the biological processes of plants responding to abiotic stresses. To clarify the possible biological functions of blueberry Dof TFs under abiotic stress, we selected eight genes with stress defense response elements in promoters. We performed qRT-PCR to analyze their expression profiles under the abiotic stress.

The results showed that the expression of VcDof genes was regulated early by salt stress (Fig. 6A). With the prolongation of stress time, VcDof1, VcDof11, and VcDof15 showed an up-regulated expression trend, and their relative expression levels were the highest at 24 h, 8.54, 3.26, and 9.07 times that of the control group, significant differences have been noticed. VcDof2 showed a down-regulated expression trend, and the relative expression level was the lowest at 24 h, which was significantly lower than the control group. Short-term drought stress also caused changes in the relative expression levels of VcDof genes (Fig. 6B). The relative expression levels of VcDof5, VcDof11, and VcDof15 at 24 h of drought stress were 7.35, 18.47, and 14.48 times higher than those in the control group. The expression trend was down-regulated at 0–12 h of stress and up-regulated at 24 h. In general, the eight blueberry Dof TFs responded positively to drought stress in the early stage and were mainly up-regulated. Under ABA stress (Fig. 6C), the relative expression levels of VcDof genes changed significantly, and VcDof1 and VcDof2 were the highest at 24 h stress, which was 9.59 and 25.28 times that of the control group. The relative expression levels of VcDof5, VcDof14, VcDof15, and VcDof49 were the highest at 3 h of stress, which were 20.40, 4.30, 5.13, and 29.26 times that of the control group. Blueberry Dof TFs responded positively to ABA stress, and the relative expression changed significantly.

Figure 6 Expression profiles of blueberry Dof TFs in response to salt, drought and abscisic acid.

(A) Expression profiles of VcDof genes under salt stress. (B) Expression profiles of VcDof genes under drought stress. (C) Expression profiles of VcDof genes under abscisic acid. Error bars indicate standard deviation, and asterisks indicate significant differences between the control and treatments, ∗p < 0.05, ∗∗p < 0.01, ∗∗∗p < 0.001.

Discussion

Identification and characterization of VcDof genes

As a class of transcription factors with C2C2-Dof zinc finger structure in plants, Dof plays an important role in plant growth and development and stress resistance (Venkatesh & Park, 2015). In this study, we have identified 51 VcDof genes in blueberry, which was lower than the number of Dof TFs in Chinese cabbage (76 BraDof genes, Ma et al., 2015). The number of Dof TFs was similar to that contained in maize (46 ZnDof genes, Chen & Cao, 2015), higher than that of Dof TFs in Arabidopsis (36 AtDof genes, Liu et al., 2020), tomato (34 SlDof genes, Cai et al., 2013), and pepper (33 CaDof genes, Wu et al., 2016). The results of multiple sequence alignment (Fig. S2) showed that all blueberry Dof TFs contained zinc-finger Dof conserved domains, and the results of gene structure and motif analysis showed that the VcDof genes in the same subfamily had similar exon or intron structure and motif ordering prove that blueberry Dof TFs are highly conserved (Wang et al., 2021a; Wang et al., 2021b).

Collinearity and duplication events analysis of VcDof genes

The Dof TFs of Arabidopsis, rice, and blueberry were constructed using MegaX to construct a phylogenetic tree, and the results showed that all Dof TFs were divided into four families (A–D) and nine subfamilies (A, B1, B2, C1, C2. 1. C2.2, C3, D1, and D2), in which the blueberry Dof gene was not found in the C3 subfamily. This is consistent with the results of studies by Wen and Lijavetzky found that cucumber CsDofs and rice OsDofs were lost in the C3 subfamily (Lijavetzky, Carbonero & Vicente-Carbajosa, 2003; Wen et al., 2016).

Gene duplication often occurs among gene family members, making gene function-specific and diverse, which is one of the main driving forces for plant genome evolution. Whole-genome duplication (WGD) promotes chromosomal recombination, gene doubling, and diversification of gene functions (Van de Peer, Maere & Meyer, 2009). Bowers’s study showed two recent WGD events occurred in Arabidopsis: the β whole-genome duplication event and the α whole-genome duplication event (Bowers et al., 2003). The amplification sources of 36 Arabidopsis Dof TFs are mainly β-genome duplication events and tandem duplication events (Wang, Tan & Paterson, 2013). The results of collinearity analysis showed that there were 24 pairs of collinearity gene pairs in 51 blueberry Dof TFs, all of which belonged to WGD. The Ka/Ks ratio of genes except VcDof7-VcDof24 and VcDof32-VcDof50 were less than 1, indicating that the WGD event of VcDof genes was the result of purifying selection. The repeated divergence events of monocotyledonous and dicotyledonous plants occurred before 170-235 Mya (Blanc & Wolfe, 2004). In this study, the divergence time of the collinearity gene of blueberry Dof TFs was before 0.2699–95.6425 Mya and later than that of monocotyledonous plants and dicotyledonous plants, which also explained why the Dof TFs of dicotyledonous plants Arabidopsis, blueberry and monocotyledonous plants rice in the evolutionary tree of this study did not have obvious segregation in clustering.

Tissue specific expression of VcDof genes

The analysis of gene expression patterns can reflect the function of genes to a certain extent. The tissue expression analysis results of this study showed that blueberry Dof TFs were differentially expressed in different tissues and developmental stages, but the overall expression was higher in the root. This is consistent with the experimental results that the Dof transcription factors were highly expressed in cucumber and pepper root tissues (Wu et al., 2016; Wen et al., 2016). The expression levels of VcDof2 and VcDof45 continued to change during blueberry flowering and fruit development. Previous studies have shown that AtDof4.1, as a transcription inhibitor, delays the flowering of Arabidopsis and inhibits the development of reproductive organs, resulting in smaller leaves, flowers, and siliques (Ahmad et al., 2013). In rice, under dark conditions, the OsDof12 (Rdd1) gene was inhibited, while the expression was up-regulated under light conditions. Over-expression of OsDof12 (Rdd1) significantly delayed the flowering time of transgenic rice under long-day conditions, and the downstream genes Hd3a and OsMADS14 were up-regulated. After interfering with OsDof12 (Rdd1) gene expression, the flowering time of rice was delayed, and rice’s grain size and thousand-grain weight were significantly reduced (Li et al., 2009). It is speculated that VcDof45 and VcDof2 play important roles in flowering regulation and fruit development of blueberries.

Potential role of VcDof genes in response to abiotic stress

Dof TFs play an important role in abiotic stress in plants. This study selected eight VcDof genes for expression pattern analysis under abiotic stress. The results showed that the relative expression levels of VcDof1, VcDof11, and VcDof15 under salt, drought, and ABA short-term induction stress were significantly higher than those in the control group, showing an upward expression trend. The VcDof gene expression differed among the three stresses, but they could respond positively to stress. According to previous studies, StDof4, StDof5, and StDof11b were up-regulated under salt and drought stress in potatoes and positively responded (Venkatesh & Park, 2015). Most BraDof genes were rapidly up-regulated under salt, drought, heat, and cold stress in Chinese cabbage (Ma et al., 2015). TaDof2, TaDof3, and TaDof6 were up-regulated, and soluble protein synthesis increased under drought stress in wheat (Liu et al., 2020). Therefore, we speculate that VcDof1, VcDof11 and VcDof15 are positive regulators in the process of blueberry resisting abiotic stress and assume important functions, but their specific regulatory mechanisms still need further study.

Conclusions

In general, 51 conserved VcDof genes were identified in the present study, distributed in eight subfamilies. Colinearity and evolution analysis showed that the main driving force of gene duplication of the VcDof was WGD, which was purify selected in the evolution process. The gene divergence event occurred after the divergence between monocotyledonous and dicotyledonous plants. The results of tissue expression analysis showed that VcDof2 and VcDof45 might play important roles in blueberry flowering and fruit development. VcDof1, VcDof11, and VcDof15 can respond positively and up-regulate expression under abiotic stress, which may play an important role in blueberry defense against abiotic stress.

Supplemental Information

Supplemental Information 1 Primer sequences used in the experiment

Click here for additional data file.

Supplemental Information 2 Raw data for qRT-PCR

Click here for additional data file.

Supplemental Information 3 The sequence used in the present research

Click here for additional data file.

Supplemental Information 4 Motifs of Dof proteins in blueberry

Click here for additional data file.

Supplemental Information 5 Multiple sequence alignment of Zinc-finger domain (marked with top) of blueberry Dof TFs proteins

Click here for additional data file.

Additional Information and Declarations

Competing Interests

Author Contributions

Data Availability

The authors declare there are no competing interests.

Tianjie Li conceived and designed the experiments, performed the experiments, analyzed the data, prepared figures and/or tables, authored or reviewed drafts of the article, and approved the final draft.

Xiaoyu Wang conceived and designed the experiments, authored or reviewed drafts of the article, and approved the final draft.

Dinakaran Elango conceived and designed the experiments, authored or reviewed drafts of the article, and approved the final draft.

Weihua Zhang performed the experiments, analyzed the data, prepared figures and/or tables, and approved the final draft.

Min Li performed the experiments, analyzed the data, prepared figures and/or tables, and approved the final draft.

Fan Zhang performed the experiments, analyzed the data, prepared figures and/or tables, and approved the final draft.

Qi Pan performed the experiments, analyzed the data, prepared figures and/or tables, and approved the final draft.

Ying Wu conceived and designed the experiments, authored or reviewed drafts of the article, and approved the final draft.

The following information was supplied regarding data availability:

The raw data is available in the Supplementary File.

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
