# Peer review of "Genome-wide identification, phylogenetic and expression pattern analysis of Dof transcription factors in blueberry (Vaccinium corymbosum L.)"

_PeerJ, doi:10.7717/peerj.14087_

## Round 0.1 · original submission · Major Revisions

Please address concerns of all reviewers and amend manuscript accordingly.

Reviewer 1 ·

Basic reporting

a. The writing should be improved to make your descriptions and statements clear and accurate. I suggest the authors have someone who is proficient in English review and help edit your manuscript. For example, lines 135-137, 145-147, 157-158, 181-182 could be rephrased to make it clear.
b. In the description of Figure 2, the authors stated that “The yellow box represented the CDS region and green box represented the UTR”, but in the figure legend CDS is in green and UTR is yellow.
c. The first part of the introduction needs more references. For example, references are needed to support your statements in lines 44-46 and 58-59.

Experimental design

a. Molecular weight and isoelectric point are important physicochemical properties of proteins, but I don’t see why it is necessary to report these values, they are not relevant to the current study. Also, what’s the purpose of averaging the molecular weight of all Dof proteins?
b. If the authors decide to keep all the data (calculated MW and pI), please include the unit when you talk about MW values, lines 180-181.
c. The method section needs more information, the current writing is unclear on how some experiments were done. For example, from the description in line 169, it’s not clear how the relative expression levels were calculated.
d. It is unclear how the statistical analysis was performed, the authors only mentioned the software used in the study (line 170).

Validity of the findings

a. In the “Potential role of VcDof genes in response to abiotic stress” section (lines 345-355), the authors didn’t discuss the potential role of these genes, only the gene up-regulation data were discussed.
b. Figure 5 has a citation (line 254), is this figure generated based on your own data or data from other researchers (the cited work)?

Reviewer 2 ·

Basic reporting

no comment

Experimental design

no comment

Validity of the findings

no comment

Additional comments

It is advised to add more detail to all figure legends.

Figure lengend_figure 2B describes that the yellow box represented the CDS region and green box represented the UTR. However, the figure is mislabeled. Please revise.

Reviewer 3 ·

Basic reporting

Authors in this manuscript studied the Dof transcription factors in blueberry and presented potential functions of blueberry Dof Tfs against abiotic stress. Li et al., identified 51 VcDof genes in blueberry and analyzed their physicochemical properties and phylogenetic relationships of the VcDof genes. Cis-acting promoter, tissue-specific, and abiotic stress expression analysis of Dof Tfs revealed that VcDof1, VcDof11, and VcDof15 play important roles in blueberry abiotic stress tolerance. Overall, the authors presented the recent findings clearly, and the discussion was written in a clear and concise manner. I support its publication given that the authors address the below issues.

Major:

1. In table 2: The Ks and Ka/Ks columns contain the same values. The authors need to correct this.
2. Line 223: The authors wrote, in line 223, “The results showed that the ka/ks ratio of 95.83% collinear gene pairs……”.
However, in table2: VcDof32- VcDof50 has Ka/Ks value of more than 1. The authors need to recheck their math. (Is the correct value be 91.66%) ?
3. Lines 225, 321: Pairs of genes were subjected to positive selection. 7&24 and 32&50 as well?
4. Line 238: The authors wrote, “12 % of VcDof genes contain an ‘O2-site’ promoter…….”. However, as per Fig 4c, it is 18 %.

Minor:

1. Line 241: correct spelling of ‘promoter’
2. Maintain consistency in writing Dof. For example, in line 178: the authors used non-italics VcDof40, and in line 182, they used italics VcDof15. In line 199, it is all caps (DOF).
3. Use space between sentences and open brackets. Example: lines 204, 207, etc
4. Lines 223, 320, and table 2: It is ‘Ka/Ks’. not ‘Ka/ks’ (capitalize K in ks)
5. Line 225: Correct the sentence. (remove ‘this’)?

Experimental design

No comment

Validity of the findings

No comment

Additional comments

No comment

---

## Round 0.2 · accepted · Accept

All concerns of the reviewers were adequately addressed and revised manuscript is acceptable now.

Reviewer 2 ·

Basic reporting

no comment

Experimental design

no comment

Validity of the findings

no comment

Additional comments

The authors have addressed my concerns, and I recommend acceptance of the manuscript.

Reviewer 3 ·

Basic reporting

The authors have addressed all of my criticisms in this revision. I recommend publishing the article in its current form.

Experimental design

No comments.

Validity of the findings

No comments.